# Nucleotides as Bacterial Second Messengers

**DOI:** 10.3390/molecules28247996

**Published:** 2023-12-07

**Authors:** Mario E. Cancino-Diaz, Claudia Guerrero-Barajas, Gabriel Betanzos-Cabrera, Juan C. Cancino-Diaz

**Affiliations:** 1Departamentos Microbiología and Inmunología, Escuela Nacional de Ciencias Biológicas, Instituto Politécnico Nacional, Manuel Carpio, Plutarco Elías Calles, Miguel Hidalgo, Ciudad de México 11350, Mexico; 2Departamento de Bioprocesos, Unidad Profesional Interdisciplinaria de Biotecnología, Instituto Politécnico Nacional, Av. Acueducto, La Laguna Ticoman, Gustavo A. Madero, Ciudad de México 07340, Mexico; cguerrerobarajas@gmail.com; 3Área Académica de Nutrición y Medicina, Instituto de Ciencias de la Salud, Universidad Autónoma del Estado de Hidalgo, Carretera Pachuca-Actopan Camino a Tilcuautla s/n, Pueblo San Juan Tilcuautla, Pachuca Hidalgo 42160, Mexico; gbetanzo@uaeh.edu.mx

**Keywords:** cyclic di-GMP, cyclic di-AMP, cyclic AMP, pppGpp, bacteria, second messengers

## Abstract

In addition to comprising monomers of nucleic acids, nucleotides have signaling functions and act as second messengers in both prokaryotic and eukaryotic cells. The most common example is cyclic AMP (cAMP). Nucleotide signaling is a focus of great interest in bacteria. Cyclic di-AMP (c-di-AMP), cAMP, and cyclic di-GMP (c-di-GMP) participate in biological events such as bacterial growth, biofilm formation, sporulation, cell differentiation, motility, and virulence. Moreover, the cyclic-di-nucleotides (c-di-nucleotides) produced in pathogenic intracellular bacteria can affect eukaryotic host cells to allow for infection. On the other hand, non-cyclic nucleotide molecules pppGpp and ppGpp are alarmones involved in regulating the bacterial response to nutritional stress; they are also considered second messengers. These second messengers can potentially be used as therapeutic agents because of their immunological functions on eukaryotic cells. In this review, the role of c-di-nucleotides and cAMP as second messengers in different bacterial processes is addressed.

## 1. Introduction

Nucleotides function to store energy from food, and they are components of nucleic acids. They are indispensable for all cells. In past decades, cell signaling by the nucleotides adenosine and guanosine was discovered to function as essential second messengers in eukaryotic and prokaryotic (bacteria and archaea) cells. In bacteria, cell signaling by nucleotides is associated with regulating biofilm formation, sporulation, cell differentiation, cell wall stability, and virulence [1]. Cell signaling nucleotides are structurally different from conventional nucleotides, mainly cyclic mononucleotides or dinucleotides, such as cyclic AMP, and others, such as cyclic di-AMP (c-di-AMP) or cyclic di-GMP (c-di-GMP) (Figure 1).

Cyclic dinucleotides are formed from conventional mononucleotides; two ATP molecules are substrates to produce c-di-AMP by specific enzymes diadenylate cyclases (DAC); in the case of c-di-GMP, GTP or GDP are the substrates, and the ATP molecules are the phosphate group donors by specific enzymes diguanylate cyclase (DGC) (Figure 2, Figure 3 and Figure 4). Similarly, non-cyclic pppGpp or ppGpp alarms are formed from ATP as phosphate donors with specific enzymes [2]. The bacterium regulates the production levels of cyclic di-nucleotides (c-di-nucleotides) for cell signaling, and the quenching of nucleotide signaling is caused by their degradation. Degradation of c-di-nucleotides occurs by c-di-nucleotide-specific phosphodiesterases (PDEs).

Sensing environmental changes, such as nutrient level, pH, osmolarity, or other stressors, drives the synthesis or degradation of c-di-nucleotides that can trigger the switching on or off of cellular processes. The c-di-nucleotides produced bind to specific bacterial proteins (receptor proteins), which perform a specific function. In this review, we will address nucleotide signaling in bacteria. First, we will discuss c-di-GMP and its variants, then cAMP and c-di-AMP, and, finally, alarmones.

### Similarities and Differences between c-di-Nucleotides

Cyclic AMP (cAMP) was initially described in glycogen metabolism in animals. In bacteria, cAMP is involved in biofilm formation, virulence, and in central metabolism [3,4]. The substantial difference between signaling in prokaryotic and eukaryotic cells is that cAMP directly binds to target proteins to function in bacteria. In contrast, cAMP requires intermediates of the protein kinase A (PKA) complex in eukaryotic cells.

The second messenger c-di-GMP subsequently was discovered. From studies of bacterial genomes, it was found that diguanylate cyclase (DGCs) enzymes have a conserved domain known as the “GGDEF” domain [5]. In the case of c-di-GMP phosphodiesterase (PDE), enzymes that degrade c-di-GMP have a highly conserved enzymatic domain known as the “EAL” domain [6]. However, a second domain with c-di-GMP degrading activity was determined in PDEs corresponding to the HD-GYP domain. In most bacteria, enzymes that participate in c-di-GMP synthesis/degradation have GGDEF, EAL, and HD-GYP domains, which are widely conserved. Other bacteria have up to dozens of enzymes associated with the synthesis/degradation of c-di-GMP [7].

The c-di-AMP was the next cyclic dinucleotide discovered [8] in several bacterial and archaeal species. Adenylate cyclases (DACs) are involved in the synthesis of c-di-AMP, and their degradation is by specific PDEs; the DAC domain is in DAC enzymes and the DHH-DHHA1 domain in PDE enzymes, degradation of c-di-AMP by PDE enzymes produces pApA [9].

There are differences between c-di-AMP and c-di-GMP. For example, one of the few genes that encode DAC enzymes is present in most bacteria. In contrast, genes for DGC enzymes are more abundant in both quantity and variety. Additionally, c-di-AMP participates in the process of bacterial growth that is essential in many bacterial species but not all [10].

Gram-positive bacteria, such as the phyla Actinobacteria and Firmicutes, frequently have DAC domain-containing proteins. On the other hand, members of phyla Fusobacteria, Bacteroidetes, Chlamydiae, Cyanobacteria, and the class Deltaproteobacteria, all Gram-negative bacteria, possess DAC domain proteins. As was mentioned, one DAC enzyme is present in most organisms. However, *Bacillus* spp. and *Clostridium* are bacteria that have two or three DAC enzymes. In contrast to c-di-GMP, most organisms can have multiple DGC enzymes. 

The c-di-AMP and c-di-GMP synthesis or degradation enzymes are distributed differently among bacteria. The phylum Proteobacteria are especially abundant in c-di-GMP synthesis enzymes. The classes Alphaproteobacteria, Betaproteobacteria, and Gammaproteobacteria have c-di-GMP synthesis enzymes but they do not possess a c-di-AMP signaling system. Instead, enzymes to synthesize both c-di-GMP and c-di-AMP are present in the class Deltaproteobacteria. In contrast, c-di-GMP-producing enzymes are absent in phyla Fusobacteria and Bacteroidetes, and in the order Chlamydiales [11]. On the other hand, *Clostridium*, *Streptomyces* spp., *Mycobacterium*, *Listeria*, and *Bacillus*, members of the phyla Actinobacteria and Firmicutes, contain both cellular signaling systems, whereas *Corynebacterium* spp., *Streptococcus*, and *Staphylococcus* have the c-di-AMP system but not the c-di-GMP system. Although the GGDEF domain is representative of the DGCs, it has been reported that the c-di-AMP synthesis or degradation enzymes in *Streptococcus* spp. and *Staphylococcus* have the degenerate GGDEF domains with the capacity to synthesize c-di-AMP or the YybT protein of *B. subtilis* that has GGDEF domain-containing PDE activity (GdpP) [12].

## 2. The c-di-GMP Nucleotide

The genomic arrangement of the EAL and GGDEF domains shows that they are most frequently present in multidomain proteins. The first protein identified with DGC and PDE activity was from *Gluconacetobacter xylinus*. This protein contains GGDEF-EAL domains arranged in tandem, indicating that it is a bifunctional protein with DGC or PDE activities [13]. The activity of these bifunctional proteins is determined by differential regulation and is dependent on the environment. At any given time, one enzyme activity will be more prevalent than the other (Figure 2).

### 2.1. Types of c-di-GMP Receptors

The c-di-GMP receptors/effectors may have the GGDEF, EAL, and HD-GYP domains. However, some domains are inactive, e.g., c-di-GMP receptors have inactive EAL domains. 

Other receptor/effector proteins with different domains including PilZ and I-site receptors. Among the receptor proteins, several transcriptional regulators are known, as well as several types of proteins with diverse functions. Non-proteins, such as riboswitches, also function as c-di-GMP receptors. 

The *Pseudomonas aeruginosa* PilZ protein, involved in pili formation, is a c-di-GMP receptor in which the PilZ domain has a high affinity for c-di-nucleotide under in vitro conditions. Other proteins possessing this domain are the BcsA protein of *G. xylinus*, the YcgR of *Escherichia coli* [14], the DgrA of *Caulobater crescentus*, and the PlzC and PlzD of *Vibrio cholerae* [15].

The *C. crescentus* response regulator PopA protein is a GGDEF-domain c-di-GMP receptor, which is involved in the cell cycle [16]. The *Myxococcus xanthus* hybrid histidine kinase SgmT and the CdgA receptor are also GGDEF-domain c-di-GMP receptors and the CdgA receptor participates in entry of predatory bacterium *Bdellovibrio bacteriovorus* to precells [17].

Similarly, proteins with EAL domains but without PDE activity can bind c-di-GMP. Examples are *P. aeruginosa* FimX, associated with motility by type IV pilus, and LapD from *Pseudomonas fluorescens* with a high affinity for binding cyclic di-nucleotide [18]. 

On the other hand, riboswitches are noncoding mRNA with specific secondary structures that can recognize molecular ligands. Upon the binding of a ligand to riboswitches, the secondary structure of the mRNA changes, leading to changes in transcription, mRNA splicing, or downstream gene translation [19]. GEMMS are riboswitches that specifically recognize c-di-GMP; the c-di-GMP-induced RNA splicing is another type of riboswitch [20].

### 2.2. Processes of c-di-GMP Signaling

Cell differentiation and biofilm formation are the first processes found in c-di-GMP signaling. Subsequently, other bacterial phenotypes were described, such as the virulence of *V. cholerae*, bacterial survival, predatory behaviors, multicellular development, the transmission of obligate intracellular pathogens, antibiotic production in streptomycetes, the transition between motile and sessile lifestyles, heterocyst formation in cyanobacteria, lipid metabolism and transport in mycobacteria, and survival to nutritional stress [21,22]. Cellular movement such as swimming, swarming, contracting, and gliding motility are mediated by c-diGMP in several Proteobacteria, Firmicutes, Spirochaetes, and Cyanobacteria [23].

### 2.3. Cell Motility

The transition from motility to non-motility involves several cellular states; first, a cell requires a surface to which to adhere temporarily and remain attached by its adhesive components. Subsequently, the cell must inhibit motility, which is where c-di-GMP comes into play. Elevated intracellular levels of c-di-GMP favor a strong counterclockwise (CCW) rotation. CCW induces smooth swimming and inhibits bacterial propagation on semisolid agar. The motility control mechanism by c-di-GMP is via the YcgR receptor, to which c-di-GMP binds to and causes flagellum quenching. This mechanism is supported because a mutation in *ycgR* gene restores bacterial motility on semisolid agar [24]. Signaling of c-di-GMP through the YcgR receptor is direct, as this receptor interacts with the flagellar motor. In addition, c-di-GMP can block the synthesis of new flagella in some bacteria, such as the case of *P. aeruginosa*. The FleQ protein (the first factor regulating flagellar gene expression) in *P. aeruginosa* is regulated in its function by c-di-GMP, which controls the expression of the flagellar regulon [25], causing flagellar gene switch-off.

### 2.4. Regulation of Biofilms

Cyclic-di-GMP mediates biofilm formation in different modalities, e.g., films in a liquid medium or on agar plates with dry, rough, and red colony morphotypes. In *P. aeruginosa*, high levels of biofilm formation correlate with high levels of c-di-GMP [26], implicating its involvement. Adhesion to an abiotic surface, the first step of biofilm formation, requires c-di-GMP induction of adhesion molecules. Furthermore, mediation by c-di-GMP regulates all components of the extracellular matrix of a biofilm, including various adhesive pili, non-fimbrial adhesins, exopolysaccharides, and extracellular DNA [27], and the control by c-di-GMP happens on transcriptional, post-transcriptional, and post-translational levels.

### 2.5. Pili as c-di-GMP Targets

Pili or fimbriae are formed by proteins that assemble to build non-flagellar appendages on the external surface of bacteria. Fimbriae have characteristics of adhesion to abiotic and biotic surfaces, which are associated with biofilm formation, and c-di-GMP is involved in the adhesion of biofilm, suggesting a possible regulation of fimbriae by this c-di-nucleotide. The type 3 fimbriae of *Klebsiella pneumoniae* helps biofilm formation on abiotic surfaces or human extracellular matrix-coated surfaces. DGC YfiN activity increases the c-di-GMP levels, which then induces transcription of the type 3 fimbriae mRNA. MrkJ PDE activity reduces the c-di-GMP levels, which down-regulates type 3 fimbriae expression. The mechanism of c-di-GMP regulation of the expression of type 3 fimbriae is by linkage of c-di-GMP to the transcriptional factor MrkH, since MrkH has a PilZ domain, making c-di-GMP-MrKH able to interact with the promoter of type 3 fimbriae and induce its expression [28].

Another type of fimbriae is the Cup, which is also a regulated by c-di-GMP. *P. aeruginosa* has five Cup types (A to E); Cups participate in biofilm formation because they modify cell adhesive properties. C-di-GMP can regulate the transcription of all Cup fimbriae except CupE. Some *P. aeruginosa* strains from cystic fibrosis patients with a variant small colony phenotype show increased YfiN and MorA DGCs activity to produce high c-di-GMP levels and expression of CupA fimbriae [29].

On the other hand, type IV pili is very diverse and ubiquitous. Type IV pili is associated with twitching mobility because these pili can polymerize and retract. Type IV pili and twitching mobility are essential for the maturation of biofilm. In *P. aeruginosa*, c-di-GMP signaling regulates type IV pili biogenesis and contractile motility [30].

### 2.6. Adhesins as a Target of c-di-GMP

In addition to pili, non-fimbrial adhesins contribute to biofilm formation. The non-fimbrial adhesive protein LapA from *Pseudomonas putida* and *P. fluorescens* participates in bacteria binding to the surface and cell–cell interaction, stabilizing the biofilm. The type I secretion system transports the LapA protein to the outside of the cell; subsequently, LapA is proteolytically processed at its N-terminus by the periplasmic protease LapG. The proteolytic activity of LapG is regulated by the transmembrane protein LapD. Low levels of c-di-GMP regulate LapD activity since LapD has a degenerate EAL domain [18].

### 2.7. Exopolysaccharide as a Target of c-di-GMP 

The polysaccharide intercellular adhesin (PIA), also known as the poly-β(1-6)-N-acetylglucosamine (PNAG) component of the biofilm matrix, is produced by a wide variety of Gram-negative and Gram-positive. In *E. coli*, *Yersinia pestis*, and *Pectinobacterium atrosepticum*, c-di-GMP activates the PNAG biosynthesis [31]. 

In *E. coli* K-12, two DGCs, DosC (YddV) and YdeH, are involved in PNAG-dependent biofilm formation, as DosC affects the transcription of the pgaABCD operon and YdeH stabilizes the transmembrane protein glycosyltransferase PgaD. The mechanism of PNAG synthesis is by binding of c-di-GMP to PgaC and PgaD proteins (both transmembrane glycosyltransferases) leading to their interaction and stimulation of glycosyltransferase activity [32]. 

Interestingly, PNAG is also produced in *Staphylococcus epidermidis* and *Staphylococcus aureus*. However, the mechanism of PNAG biosynthesis is independent of c-di-GMP. In fact, in staphylococci there is no c-di-GMP signaling, since only proteins with functional GGDEF domains (GdpS) are present [33].

### 2.8. Alginate Polysaccharides, Pel, and Psl as Targets of c-di-GMP

Cyclic-AMP is involved in the production of exopolysaccharides alginate, Pel, and Psl. Although several bacteria are known to produce these exopolysaccharides, they are studied most extensively in *P. aeruginosa*; the non-mucoid biofilms of this bacterium have an extracellular matrix of Pel and Psl polysaccharides. While the genetic capacity to produce Pel polysaccharides appears ubiquitous among all *P. aeruginosa* strains, not all *P. aeruginosa* strains can produce Pel polysaccharides. Strains which fail to produce Pel lack a functional the *psl* operon. C-di-GMP positively induces the transcription of the exopolysaccharides Pel and Psl by the REC-GGDEF-DGC response regulator WspR [34].

The transcription of proteins which synthesize Pel and Psl is controlled by FleQ, a c-di-GMP-dependent transcriptional regulator and receptor. In the absence of c-di-GMP, FleQ forms a complex with the ATP-binding accessory protein FleN and binds to sites both upstream and downstream of the pel Pel promoter to inhibit transcription. In the presence of c-di-GMP, FleQ binds to c-di-GMP and separates from the ATP-binding accessory protein FleN, triggering the transcriptional activity of Pel [35]. In addition, feedback exists between c-di-GMP and Psl, as Psl can elevate c-di-GMP levels via the SiaD and SadC DGCs [36]. 

Finally, *P. aeruginosa* mucoid strains also have exopolysaccharide alginate. The polymerization or transport of alginate is by Alg44 (PA3542) protein; this protein has a PilZ domain and is regulated by c-di-GMP [37].

C-di-GMP controls biofilm formation at the transcriptional level. For example, seven DGCs and four PDEs regulate the rough colony formation in *V. cholerae*. Expression of Vibrio polysaccharide (VPS) confers a rough colony. The positive transcriptional regulators VpsR and VpsT activate the expression of Vps genes. These transcriptional factors, VpsT and VpsR, are c-di-GMP receptors, and when the c-di-nucleotide binds to them, they are activated for transcription of Vps polysaccharides [38].

Furthermore, low levels of c-di-GMP correlate with low amounts of biofilm. For example, extracellular DNA (eDNA) produced by cell lysis is induced by low levels of c-di-GMP, and eDNA is necessary for mature biofilm structures [39]. Another action of low levels of c-di-GMP occurs in motile cells on the biofilm surface, which become sessile cells at low levels of c-di-GMP.

### 2.9. Cyclic di-GMP and Quorum Sensing

Quorum sensing can negatively regulate biofilm formation, and c-di-GMP induces biofilm formation. In *V. cholerae* biotype El Tor, high cell density levels produce a high concentration of the quorum-sensing autoinducer, triggering the expression of HapR, the primary regulator of quorum sensing. HapR regulates the expression of at least fourteen of the fifty-two GGDEF/EAL domain proteins and four HD-GYP domain proteins, leading to an overall reduction of the intracellular concentration of c-di-GMP [40] and thus inhibiting biofilm formation.

### 2.10. Cellulose Biosynthesis

Cellulose biosynthesis in *G. xylinus* was the first process discovered in bacteria to be regulated by c-di-GMP. The cellulose synthase enzymes have a PilZ domain at the C-terminal end, suggesting that these enzymes can be allosterically regulated by c-di-GMP. Indeed, it has been observed that the PilZ domain of *G. xylinus* cellulose synthase can join c-di-GMP, increasing the activity of this enzyme. In vitro, cellulose biosynthesis has been shown to require c-di-GMP, UDP-glucose, and membrane fractions of *G. xylinus* [41].

### 2.11. Cell Cycle Regulation and Differentiation

The freshwater bacterium *C. crescentus* undergoes morphological development from a free-swimming swarmer to a sessile stalked cell. While the swarmer cell possesses polar flagella and adhesive pili at each cell pole, an adhesive holdfast and a stalk are subsequently developed over the same pole to promote surface adhesion. This cellular differentiation within a stalked cell is strongly coupled with the initiation of the cell cycle.

The stalked cell becomes competent for DNA replication and subsequently initiates an asymmetric cell division to produce a swarmer cell. After cell division, the stalked cell begins a new round of replication. In contrast, the swarmer cell has stopped replication unless the cell attaches to a surface and begins to develop into a stalked cell. Consistent with its positive role in promoting sessility, peak levels of c-di-GMP occur during the transition from a swarmer cell to a stalked cell [42]. Collectively, two diguanylate cyclases DGCs, PleD and DgcB, drive the differentiation of the swarmer cell to a stalked cell through holdfast biogenesis and stalk initiation and elongation [43], which appear to be the proteins responsible for c-di-GMP synthesis.

### 2.12. Cell Differentiation in Multicellular Bacteria

C-di-GMP is involved in cell differentiation and multicellular bacteria. Under conditions of nitrogen starvation, the cyanobacterium *Anabaena* forms heterocysts, specialized cells with the function of carrying out oxygen-sensitive nitrogen fixation; the heterocysts supplement nitrogen to the multicellular filaments. This process of heterocyst formation is associated with c-di-GMP because the inactivation of the All2874 protein, which is a GGDEF domain DGC, inhibits this process [44].

In the case of multicellular bacteria, *Streptomyces coelicolor* starts its life cycle from the germination of a spore to produce a branched vegetative filament, which eventually forms a network of multinucleate aerial hyphae (mycelium). Overexpression of CdgA, a DGC enzyme, blocks aerial hyphae formation and generates a bald colony phenotype. The c-di-GMP also affects mycelium formation in *S. coelicolor*. Pigmentation and production of the antibiotic actinorhodin are also affected [45].

### 2.13. Virulence

The c-di-GMP signaling pathways can affect various aspects of bacterial virulence. In *V. cholerae*, the first evidence of the role of c-di-GMP in virulence was that low levels of c-di-GMP induce elevated cholera toxin expression in vitro, while high levels of c-di-GMP attenuate virulence in murine cholera [46]. These observations indicate that low levels of c-di-GMP are required for acute infection. Recent studies support this view since pathogenic bacteria have few DGC proteins, and eliminating the DGCs function demonstrates that c-di-GMP signaling is not necessary for virulence in *Yersinia pestis* [21]. The infection of spirochete *Borrelia burgdorferi* in mice causing Lyme disease occurs when the bacterium has deleted its single DGC [47]. *Salmonella enterica* serovar Enteritidis, with the elimination of all GGDEF domain genes, reduces its infectivity in mice [48]. Similar observations are in *Brucella melitensis* [49]. In addition, high levels of c-di-GMP inhibit acute infection of *B. burgdorferi*, *Y. pestis*, and *Francisella novicida* [50].

### 2.14. Cyclic di-GMP as an Immunomodulator

Cyclic-di-GMP inhibits cell proliferation in the human acute lymphoblastic leukemia cell line, the human CD4 T-lymphoblast cell line Jurkat [51], and human colon cancer cells [52]. The inhibition of cell proliferation is explained by the binding of c-di-GMP to the p21ras protein (member of the Ras superfamily of GTPases) [51], suggesting that the c-di-nucleotide has antitumor effects.

Cyclic-di-GMP is an immunostimulator that reduces the effects of *Staphylococcus aureus* infection and bacterial pneumonia caused by *Streptococcus pneumoniae* or *Klebsiella pneumoniae* in mice [53]. C-di-GMP can recruit neutrophils, monocytes, and granulocytes. The application of c-di-GMP as an adjuvant enhances the immune response against mutant staphylococcal enterotoxin, pneumococcal surface protein A (PspA), *Staphylococcus aureus* agglutination factor A (ClfA), and pneumolysin toxoid antigens. These observations demonstrate the potential use of c-di-GMP as an immunotherapeutic agent. 

Finally, the immunological effects of c-di-GMP are attributable to its binding to the cytoplasmic domain of the transmembrane protein STING [54]; c-di-GMP-STING binding activates TANK-binding kinase 1 (TBK1), and this kinase activates interferon regulatory transcription factor 3 (IRF3), resulting in the production of type I interferons [54].

## 3. The Novel Cyclic Dinucleotide Second Messengers

There are other types of signal nucleotides with structures different from those mentioned above, such as cyclic AMP-GMP and cyclic AMP-AMP, and cyclic trinucleotides, such as cyclic AMP-AMP-GMP (cAAG) and cyclic AMP-AMP-AMP (cAAA). These nucleotides are synthesized by a new superfamily of proteins called cGAS/DncV-like nucleotidyltransferases (CD-NTases), and these CD-NTases are conserved in all bacteria observed. The signaling of these nucleotides can activate effector proteins, such as DNA endonucleases, transmembrane pore-forming proteins, patatin-like phospholipases, and proteases. These effector proteins provide antiphage immunity [55], protecting bacteria against bacteriophage infection.

The hybrid dinucleotide between AMP and GMP is cyclic AMP-GMP. Cyclic AMP-GMP is produced by the CD-NTase DncV [56], and the homologous DncV is present in a subset of proteobacteria, including the diarrhoeagenic *E. coli* strain DEC8D. In *Vibrio cholerae* El Tor, the dncV gene is located in the pandemic island 1 (Vsp-1) of vibrio, which is believed to have contributed to the biotype that caused pandemics. Elimination of *dncV* gene in *V. cholerae* inhibits intestinal colonization in a mouse model because DncV represses chemotaxis.

Another system of bacterial protection against nucleic acid entry events is the CRISPR-Cas system; which is considered a mechanism of immunity. The CRISPR-Cas type III system destroys transcription-dependent DNA. A component protein of the system, Cas10, is considered to be a nuclease that cleaves single-stranded DNA. During the process of cleavage of foreign DNA by Cas10, this protein also synthesizes a cyclic oligoadenylate (cOA, 4 adenine units). COA binds to Csm6/Csx1 proteins with ribonuclease activity (also belonging to the CRISPR-Cas system) to degrade RNA non-specifically; it can also bind to other RNases, such as RelE and PIN, and to DNAases to cleave RNA and bacterial DNA. COA can be toxic to bacteria, so it is degraded by CRISPR-associated nuclease 1 (Crn1), which cleaves cOA specifically into di-adenylate products in a linear fashion to quench cOA-activated effector proteins [57].

## 4. The cAMP Nucleotide

Cyclic AMP (cAMP) nucleotide signaling was first discovered in animal cells and is considered the most abundant cyclic nucleotide used by organisms. cAMP is synthesized by adenylyl cyclases (ACs) from ATP and is degraded by phosphodiesterases (PDEs; Figure 3). There are six classes of cAMP enzymes; class III is a large group of cAMP enzymes and the most diverse of all classes, and within this class are most bacterial cAMP enzymes [58].

### 4.1. cAMP Binds to cAMP Receptor Protein (CRP)

Signaling by cAMP in bacteria is mainly at the level of regulation of gene expression, whereas in eukaryotic cells, the mechanism is different. The cAMP receptor protein (CRP) family functions as a transcription factor in bacteria, and cAMP binds to CRP directly in order to activate it. These receptors counteract different cellular events directly whereas, in eukaryotic cells, cAMP signaling is first mediated by an intermediate, commonly the protein kinase A (PKA) complex, and, subsequently, the activation of transcription factors [59] and one of the functions of cAMP in eukaryotic cells was in hormonal signal transduction. 

In bacteria, cAMP regulates the glucose response or catabolic repression [60]. Low glucose levels reduce cAMP production, activating the lac operon by binding cAMP to the Crp protein to form the cAMP–Crp complex. In recent years, the study of cAMP in bacteria has discovered other processes mediated by this messenger, such as bacterial virulence, toxin control, and activation of master regulators.

### 4.2. cAMP Signaling in Pathogenic Bacteria

In pathogenic bacteria, cAMP signaling is essential because cAMP regulates biofilm formation, the type III secretion system, carbon metabolism, and the regulation of virulence genes. Intracellular pathogenic bacteria control or modulate cAMP levels within their host cells [61] to carry out their infection process.

Glucose plays an essential role in activating the AC enzyme; glucose depletion leads to the induction of AC enzyme activity in some bacteria but not others. Although the primary mechanism of cAMP signaling is via CRP family transcription factors, other diverse mechanisms of cAMP signaling within bacteria (other cAMP-associated regulatory targets) are possible and sometimes amplified by cAMP-mediated coregulation of other global regulators [62]. Some examples of cAMP signaling in pathogenic bacteria are mentioned below.

### 4.3. Pseudomonas aeruginosa

*P. aeruginosa* can express three ACs: CyaA, CyaB, and ExoY, indicating that cAMP plays a role in the pathogenesis of this bacterium. ExoY is an AC toxin that is activated within host cells, and this toxin is exported and introduced into the host cell via a cAMP-regulated T3SS secretion system [63]. In contrast, the enzymes CyaA and CyaB are not exportable and remain within the cytosol of *P. aeruginosa*; these intracellular ACs increase intracellular cAMP levels to control gene expression of virulence proteins. Ca^2+^ is an environmental component for the expression of the T3SS secretion system; low Ca^2+^ concentration triggers cAMP production and, as a result, the expression of secretion system components [64]. Ca^2+^ is sensed in the environment by CyaB and detecting a low Ca^2+^ concentration activates CyaB to produce high intracellular cAMP concentrations [64]. Furthermore, CyaB is involved in virulence, deletion of the *cyaB* gene attenuates this process [65]. Low virulence is a result of low expression of T3SS components. Overexpression of the regulatory factor, ExsA, restores the virulence phenotype because of the reconstitution of the expression of the T3SS component system [65]. Another cAMP-activated transcription factor is Vrf. Cyclic-AMP binding to Vrf regulates transcriptional upregulation of many virulence-associated genes, such as type IV pili, the T3SS secretion system, exotoxin A, and the quorum sensing system [66]. On the other hand, cAMP degradation is driven by the CpdA protein to reduce the amount of intracellular cAMP, and these conditions are necessary to overexpress virulence factors.

### 4.4. Vibrio cholerae

Cyclic-AMP signaling is inhibited by *V. cholerae* toxin (CT) during infection of intestinal epithelial cells, causing diarrhea. However, cAMP signaling in *V. cholerae* has essential functions for the bacterium, such as integrating the available carbon source, promoting biofilm formation, sensitivity to bacteriophages, and expressing virulence genes. For example, in *Vibrio* spp., a single AC enzyme, CyaA, produces cAMP, with signaling via the transcription factor CRP [67]. Glucose is an essential component of the cAMP signaling process. Low glucose levels increase cAMP production by CyaA and, subsequently, cAMP binds to the Crp transcription factor. The cAMP-Crp can regulate biofilm formation, quorum sensing, toxin-regulated pilus (TCP), and CT expression [68].

### 4.5. Mycobacterium tuberculosis

Unlike other pathogenic bacteria, *Mycobacterium tuberculosis* is distinguished by high levels of cAMP signaling. For example, *M. tuberculosis* has 16 AC-like proteins and 10 confirmed AC, indicating that cAMP function is essential in this bacterium. Another difference is that low levels of glucose do not produce a significant effect on cAMP levels. In contrast, the AC activity is controlled by various host conditions; for example, fatty acids, carbon dioxide (CO_2_), and pH affects the activity of *M. tuberculosis* ACs, while hypoxia and starvation affect the expression of genes encoding ACs [61]. *M. tuberculosis* within the macrophage is known to produce and secrete cAMP. Production is by the AC, Rv0386, which has been corroborated by deleting the corresponding gene of Rv0386, and results in a phenotype of decreased virulence and pathology of *M. tuberculosis* in a murine infection model [69].

In addition to the different ACs enzymes of *M. tuberculosis*, ten putative cAMP-binding proteins have been identified as receptors. Among them are two transcription factors of the CRP family and a protein lysine acetylase. Of the two transcription factors, one Crp induces the expression of a regulon, where it simultaneously controls the expression of about 100 genes. An *M. tuberculosis* strain with Crp deletion significantly reduces its virulence in a murine model [70]. The second Crp transcription factor, Cmr, is involved in macrophage response and in the expression of genes in response to cAMP levels [71].

## 5. The c-di-AMP Nucleotide

The c-di-AMP dinucleotide was initially come across bound to the DNA integrity scanning protein (DisA) of the bacterium *Thermotoga maritima* [72]. Subsequently, intracellular pathogenic bacteria (such as *Listeria monocytogenes*) were observed to secrete c-di-AMP to the cytosol of infected host cells [73]. The bacteria *Bacillus subtilis* [74], *Chlamydia trachomatis* [75], *Staphylococcus aureus* [76], and *Streptococcus pyogenes* [77] produce c-di-AMP.

Like the other nucleotides, c-di-AMP binds to a specific set of receptor or target proteins; this binding modifies the functions of these proteins or may influence other effector proteins regulated by the receptor, thereby controlling cellular pathways. The c-di-AMP participates in several processes, including bacterial growth under low potassium conditions [78], regulation of fatty acid synthesis in *Mycobacterium smegmatis* [79], cell wall homeostasis [80], and sensing of DNA integrity in *B. subtilis* [81] (Figure 4).

### 5.1. c-di-AMP Receptors

The first c-di-AMP receptor was the Ms5346 protein from *Mycobacterium smegmatis* and corresponds to a regulator of the tetracycline resistance family (TetR) that was later renamed the c-di-AMP receptor regulator (DarR). In *M. smegmatis*, DarR binds to its own promoter and thus represses its transcription. DarR binds to the promoters of the medium-chain acyl-CoA ligase transcription operon and the family of prominent transporter facilitators. The c-di-AMP–DarR complex causes increased interactions with DNA [79], so changes in the level of c-di-AMP affects fatty acid synthesis in *M. smegmatis*.

In *S. aureus*, KtrA, KdpD, PstA, CpaA, and OpuCA are c-di-AMP target proteins [82]. CpaA is a predicted cation/proton antiporter protein, and OpuCA is the ATPase component of the ATP-to-carnitine binding cassette transporter OpuC [83]. 

The KtrA protein is a member of the Ktr-type potassium transport system [1]. KtraA binds with KtrB-type membrane components to form a potassium transport system, which has been detected in various bacteria [84]. The c-di-AMP binds to the KtrA protein, causing a conformational change in the transporter components that ultimately make the KtrB protein able to open or close the potassium transporter channel. In addition, bacteria mutants in *gdpP* gene, which produce high levels of c-di-AMP, are sensitive to salt stress and, similar to ktrA mutant strains, require high potassium levels to grow under osmotic stress conditions [78].

The c-di-AMP binds to the histidine kinase protein KdpD to control its function [85]; the KdpD protein is involved in potassium homeostasis [86]. KdpD protein and KdpE induce the expression of a second type of potassium uptake system [87].

Another c-di-AMP receptor protein is PstA, which responds to cells’ nitrogen and carbon status by sensing glutamine and 2-ketoglutarate levels [73,88].

### 5.2. Cellular Processes Regulated by c-di-AMP

#### 5.2.1. Cell Wall Homeostasis

Changes in c-di-AMP levels compromise bacterial cell wall integrity. In *S. aureus*, DacA and GdpP enzymes synthesize and degrade c-di-AMP, respectively. The involvement of c-di-AMP on the cell wall has been studied with GdpP-deficient strains, with a consequent an increase in the intracellular levels of c-di-AMP. The resulting phenotype have bacterial growth retardation and increased resistance to acid stress and beta-lactam antibiotics in *S. aureus* [89]. These mutants have increased peptidoglycan cross-linking. By microscopy, the GdpP-deficient *S. aureus* strain significantly reduces cell size compared to the wild type [76].

In contrast, attempts to obtain *dacA* gene mutants in *S. aureus* have failed, suggesting that c-di-AMP production is vital for growth in this bacterium. Strains of *B. subtilis* and *L. monocytogenes* that have a mutated *gdpP* gene display increased resistance against beta-lactam antibiotics. In the case of *Lactococcus lactis* and *L. monocytogenes.* GdpP-mutant strains are more resistant to heat stress [90] and acid [91], all indicating that cell wall changes occur in strains with high levels of c-di-AMP [76]. In contrast to GdpP-mutant strains, strains overexpressing this protein have decreases in the level of c-di-AMP, and in *B. subtilis* and *L. monocytogenes*, it has been observed that they are susceptible to antibiotics targeting the cell wall [91].

#### 5.2.2. Sensing of DNA Damage

When DNA is damaged, detection and repair mechanisms are activated. DisA is a sensor protein that can scan the chromosome to detect alterations on DNA; when DisA performs this function, it simultaneously synthesizes c-di-AMP [73]. The DisA movement on DNA depends on the synthesis of c-di-AMP [73]. For example, when DNA strands are at a stalled replication fork or Holliday junction, c-di-AMP synthesis and DisA movement stop, which decreases the level of c-di-AMP. This decrease in c-di-AMP levels is a likely signal to stop replication and initiate the DNA repair mechanism by the RadA protein. In the case of bacterial sporulation, DNA damage is detected by DisA, resulting in the inhibition of sporulation initiation. Under these circumstances, c-di-AMP levels fall, and restoration of sporulation initiation can be triggered by the exogenous addition of c-di-AMP [74], suggesting that c-di-GMP levels are a signal to control DNA damage. Furthermore, it has been observed that in *B. subtilis* cells, the GdpP expression increases when DNA is exposed to an agent that damages this molecule, resulting in low c-di-AMP levels and sporulation rate [74]. Therefore, c-di-AMP levels indicate DNA integrity and help ensure that only undamaged DNA is packaged within the *B. subtilis* spore.

### 5.3. c-di-AMP in Eukaryotic Host Cell

C-di-nucleotides control bacterial physiology. However, c-di-AMP can also be detected by eukaryotic cells. Intra- and extracellular pathogens release c-di-AMP, and this c-di-nucleotide stimulates the immune response of eukaryotic cells, principally in the production of type I IFN. Intracellular bacteria that escape from the vacuolar compartment and propagate into the cytosol release c-di-AMP, through the multidrug efflux pumps MdrM and MdrT. The presence of c-di-AMP in the cell cytosol drives a response dependent on the helicase DDX41 and the transmembrane receptor STING [54,92]. When c-di-AMP activates STING, it interacts with TBK1 kinase to activate it [93]. This kinase activates the transcription factor IFN regulatory factor 3 (IRF3). Activated IRF3 translocates to the nucleus and induces the type I IFN production. The c-di-AMP secreted by pathogenic bacteria can lead to immune recognition and modulation, which is proposed for therapeutic use as vaccine adjuvants, highlighting another potential use of these c-di-GMP and c-di-AMP molecules [52].

## 6. The pppGpp or ppGpp Nucleotide

Nutrients become limiting during bacterial growth, mainly in the stationary phase; in this condition, the bacterium resorts to other processes to reuse possible nutritional sources within the cell. The processes of synthesis of membrane components, ribosomal proteins, RNA, and DNA are arrested, and in response, the cell rapidly produces factors crucial for stress resistance [94]. In general, these events are known as nutritional stress responses. Under nutritional stress conditions, a signaling process is triggered by the nucleotide’s guanosine tetraphosphate and guanosine pentaphosphate (ppGpp and pppGpp, respectively), called alarmones.

Two classes of enzymes carry out the synthesis of alarmones. One is the monofunctional synthetase enzyme, and the other is the bifunctional synthetase-hydrolase enzyme. The monofunctional synthetase corresponds to the RelA protein; this enzyme uses GTP and ATP as phosphate donors to generate ppGpp and then convert it into pppGpp. The bifunctional synthetase-hydrolase enzymes are represented by SpoT, Rel, or RSH (Rel-Spo homologous) proteins. These enzymes synthesize ppGpp or pppGpp and hydrolyze these nucleotides to produce GDP and pyrophosphate (PPi) or GTP and PPi, respectively.

These two classes of enzymes participate in alarmone synthesis by different metabolic states. RelA functions in alarmone synthesis when there is a shortage of amino acids and this causes uncharged tRNAs, which stimulate this enzyme’s activity. In contrast, the bifunctional synthetase-hydrolase enzyme is involved when bacteria are under nutritional stress from carbon, fatty acids, iron, or phosphate starvation. The synthesis of ppGpp under nutritional stress conditions causes a controlling effect on RNA polymerase (RNAP) activity in cooperation with the suppressor DnaK (DksA). The binding of ppGpp and DksA regulates transcription positive or negative by RNAP, as determined by the intrinsic properties of the promoter in question and causes numerous physiological effects [95].

Notably, DksA and ppGpp also regulate RNAP transcription through an event called σ-factor competition. This event occurs as follows: in the logarithmic growth phase, vegetative σ-factor 70 (RpoD) binds to RNAP to initiate DNA replication and lipid synthesis as well as transcription of essential proteins. In stringent conditions, bacteria produce high concentrations of ppGpp to inhibit RNAP-σ70 binding; consequently, RNAP is free to bind to other types of sigma factors, such as sigma E and sigma S, which are present when the bacterium is under stress conditions [96]. When metabolic precursors are restored to high concentrations, SpoT degrades ppGpp. Thus, the σ-vegetative factor, σ70, controls RNAP to transcribe genes crucial for DNA replication and biomolecule synthesis.

A mechanism is now known in *E. coli* to stabilize and activate sigma S function under stress conditions. In the logarithmic growth phase, the sigma S protein is bound to the RssB protein (an adaptor protein), and this sigma S-RssB binding drives the proteolytic degradation of sigma S by the ClpXP proteasome [97]. However, during phosphate stress, the hydrolase activity of SpoT is inhibited. Therefore, ppGpp synthesis is elevated, then ppGpp binds to RNAP to transcribe and produce IraP and IraD, which are anti-adaptor proteins that sequester RssB and prevent sigma S delivery to the ClpXP proteasome [98].

When the concentration of amino acids is low, the bacterium uses the ppGpp message to recover amino acids from conventional proteins, such as ribosomal proteins. In *E. coli*, the ppGpp alarmone inhibits the transcription of ribosomal protein-coding genes, and the alarmones inhibit exopolyphosphatase activity, causing accumulation of polyphosphate in the cytosol. Accumulated polyphosphate binds to free ribosomal proteins which are subsequently degraded by Lon protease, producing free amino acids for enzymatic biosynthesis [99].

Under stress conditions, amino acid degradation to produce by products of metabolic interest are regulated by alarmones. In *E. coli*, ppGpp inhibits lysine decarboxylase when amino acids are limited; ppGpp interacts with LdcI (lysine decarboxylase inducible) protein to inhibit its activity, thus cytoplasmic lysine is preserved for protein synthesis. Similarly, protein synthesis and DNA replication are down-regulated by ppGpp through direct inhibition of translation elongation factor activity and DNA primase, respectively.

Alarmones combine the nutritional stress response with the virulence of pathogenic bacteria [100]. Within the phagocyte, *Legionella pneumophila* produces ppGpp to induce activation of two (noncoding) regulatory RNAs, RsmZ and RsmY. In the stationary growth phase, RsmZ and RsmY activate the sigma S factor and the two-component LetA-LetS system [101]. After intracellular infection, *Salmonella* spp. Typhimurium induces the PhoPQ two-component system. The PhoPQ system expresses SlyA protein, and SlyA regulates the transcription of genes essential for virulence in *S. typhimurium* [102]. The ppGpp messenger activates the function of the SlyA protein, facilitating the binding of SlyA to target promoters involved in the virulence of this bacterium [103].

### Interactions between ppGpp, c-di-GMP, and c-di-AMP

In contrast to the inhibition of degradation, signaling interactions between ppGpp and c-di-GMP promotes biofilm formation. Low doses of antibiotics which act to inhibit of translation, such as erythromycin, tetracycline, or chloramphenicol, induce degradation of ppGppp by SpoT activity and lead to an increase in the expression of *pgaA* gene, which encodes the protein necessary to synthesize poly-acetylglucoseamine (poly-GlcNAc) [31]. Poly-GlcNAc mediates biofilm formation [reference]. Conversely, maximal poly-GlcNAc production also requires the formation of c-di-GMP via the enzyme DGC YdeH [31], suggesting that bacteria in antibiotic stress inactive alarmone signaling to induce a barrier of protection (biofilm) against antimicrobial agents.

C-di-AMP likely influences stringent signaling; in vitro, ppGpp inhibits phosphodiesterase activity to which degrades c-di-AMP [100]. Consequently, ppGpp maintains the cellular pool of c-di-AMP. The interaction between ppGpp-mediated signaling and c-di-nucleotides shows the influence of alarmones on bacterial physiology.

## 7. Clinical Application of di-Nucleotides

Drug design to inhibit c-di-nucleotide signaling is an alternative to control bacterial biofilm formation. The approach is to use small molecules that act on the c-di-AMP-mediated transition from planktonic to sessile lifestyles [104]. Decreased biofilm formation and virulence factors were achieved by inhibiting c-di-GMP synthesis; for example, N-4-anilinophenyl-benzamide (DI-3) reduces c-di-GMP synthesis in *P. aeruginosa* and *V. cholerae*. Similarly, N-2-henylethyl-aminocarbonoyl-benzamide (DI-8), inhibits *V. cholerae* [105]. Another mechanism is the global reduction of c-di-GMP levels, for which azathioprine, 6-mercaptopurine, and terrein are effective on *E. coli* and *P. aeruginosa* [106,107,108]. Other molecules affect di-nucleotide signaling by competition with c-di-GMP to binding or sequestration sites are thiol-benzo-triazole-quinazolinones, ebselen, and the proline-rich tetrapeptide Gup-Gup-Nap-Arg; these molecules were tested on *P. aeruginosa* [109,110,111].

Another approach is to inhibit enzymes that synthesize or degrade c-di-GMP; raffinose inhibits biofilm matrix formation by activating c-di-GMP-specific PDE to reduce the level of c-di-GMP [112]; inhibitors of DGC are D-di-GMP and GTP; an anti-c-di-GMP peptide binds to the DGC YfiN of *P. aeruginosa*, which impacts the signaling function of c-di-GMP, interfering with motility and biofilm formation [113].

Cyclic-di-AMP has been used to modulate the immune response. The c-di-AMP worked as an adjuvant to promote an immune response to against the influenza virus [114]; this strategy also worked against the hepatitis C virus [115] and simian immunodeficiency virus [116]. Regarding vaccines, c-di-AMP has been used for vaccination against tuberculosis [117] and *Trypanosoma cruzi* [118]. An oral vaccine prepared with the vaccine antigen and c-di-AMP with the probiotic lactobacillus has also been used [119].

## 8. Conclusions and Prospects

Although c-di-nucleotides as second messengers are heavily involved in various bacterial biological processes, the molecular mechanism of the environmental signal that triggers their synthesis or degradation has yet to be elucidated. Moreover, the processes involving c-di-nucleotides are complex, and, as mentioned above, c-di-nucleotides bind to receptor proteins as key transcription factors that regulate these processes. So far, the inactivation of c-di-nucleotide signaling when the biological process has completed its function has yet to be discovered in detail. Other biological processes may exist in which c-di-nucleotide signaling and other receptor proteins yet to be discovered are involved. In addition, c-di-nucleotide signaling may be directed to inhibit bacterial biological processes, especially virulence factors in pathogenic bacteria, to control or inhibit bacterial infections. New approaches are needed to use these c-di-nucleotides as therapeutics for eukaryotic cells or to treat intracellular pathogenic bacteria.

## Figures and Tables

**Figure 1 molecules-28-07996-f001:**
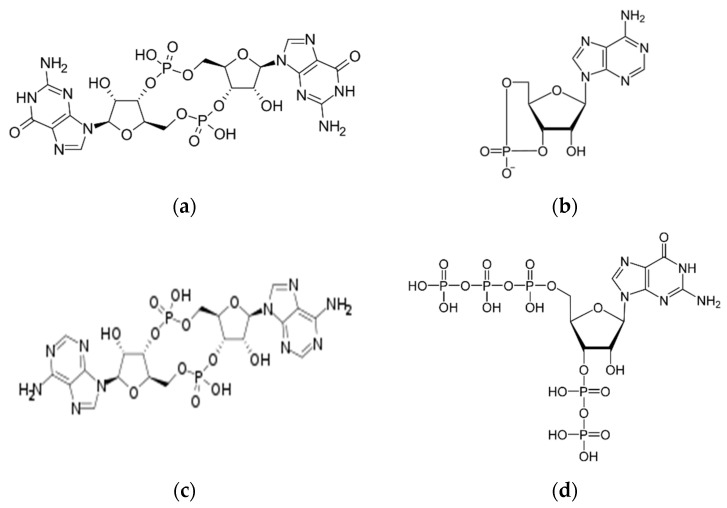
Chemical structures of nucleotides. Nucleotides involved in cell signaling. (**a**) Cyclic diguanylate (c-di-GMP), (**b**) Cyclic adenosine monophosphate (cAMP), (**c**) Cyclic-di-adenosine monophosphate (c-di-AMP), (**d**) Guanosine pentaphosphate (pppGpp).

**Figure 2 molecules-28-07996-f002:**
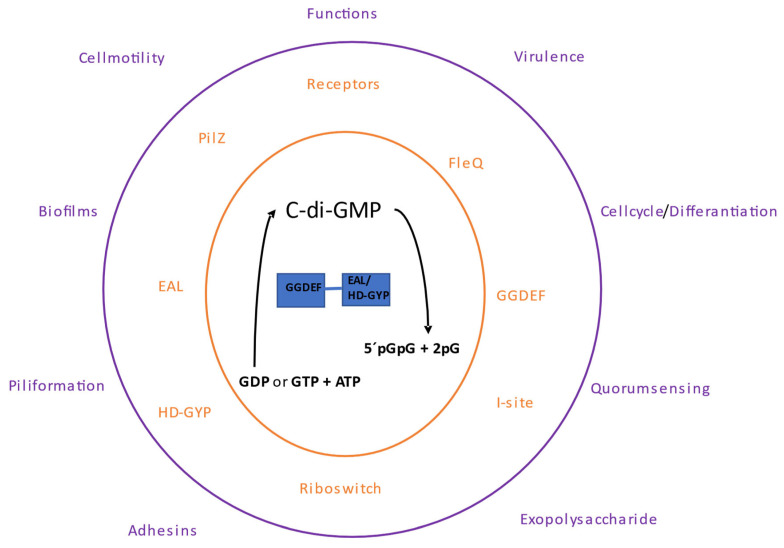
Signaling by c-di-GMP. The center shows the synthesis and degradation of c-di-GMP; the GGDEF domain is involved in the synthesis, and the EAL or HD-GYP domains are involved in its degradation. In the central orange part, the dinucleotide receptor proteins are shown. In the outer purple part, the biological processes in which c-di-GMP participates are shown.

**Figure 3 molecules-28-07996-f003:**
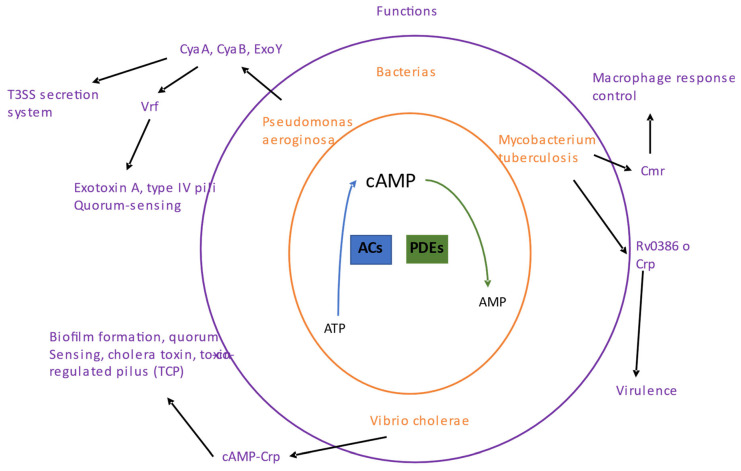
Signaling by cAMP. The center shows the synthesis and degradation of cAMP. Adenylate cyclase enzymes (ACs) are involved in synthesizing c-di-AMP and specific phosphodiesterases (PDEs) in their degradation. Outside the center, in orange, some pathogenic bacteria with cAMP functions are shown. The outer part in purple shows the biological processes where c-di-AMP is involved for each bacterium.

**Figure 4 molecules-28-07996-f004:**
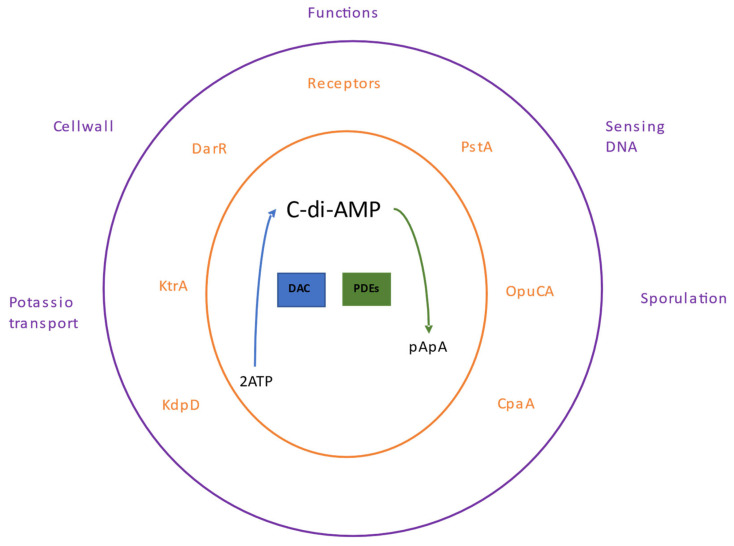
Signaling by c-di-AMP. The synthesis and degradation of c-di-AMP is shown in the center. Enzymes with DAC domains participate in the synthesis of c-di-AMP and specific phosphodiesterases (PDEs) in their degradation. Dinucleotide receptor proteins are shown in the central orange part. The biological processes where c-di-AMP is involved are shown in the outer purple part.

## Data Availability

Data are contained within the article.

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
