# Peer review of "Nucleotides as Bacterial Second Messengers"

_molecules, 2023, doi:10.3390/molecules28247996_

Round 1
Reviewer 1 Report
Comments and Suggestions for Authors
Thank you for submitting your review to molecules
Here are my few comments
The review article talked about the significance of nucleotides as secondary messengers within bacteria. The authors started with an overview of c-di-GMP functions concerning biofilm regulation and cell motility, highlighting its downstream effects on pili, adhesions, and polysaccharides. Then, they explained the distinct roles of cAMP signaling within various pathogenic bacteria and the contribution of c-di-AMP in recognizing DNA damage. Finally, The authors disclosed the interaction between alarmones and c-di-AMP, along with c-di-CMP.
Although this constitutes a review article and not a research-based manuscript containing results and discussions, it's still valuable. Prior literature, such as the articles found at the two provided links, have indeed explored the same title. Nevertheless, the addition of varied review articles within the same area of study continues to enrich the academic literature.
https://www.sciencedirect.com/science/article/pii/S136952740900006X
https://www.sciencedirect.com/science/article/pii/S0022283619300129
This manuscript simplifed some of the roles of nucleotides as secondary messengers in bacteria however, the authors were unable to elucidate the molecular mechanism of the environmental signal that triggers their synthesis and degradation.
the authors were unable to elucidate the molecular mechanism of the environmental signal that triggers the synthesis and degradation of these secondary messengers which is still a gap in the literature but yes, I can say that their conclusion is consistent with the argument presented.
As I mentioned in my first comment, having more than 100 references is too much, however the authors tried to avoid self-citation.
Comments on the Quality of English Language
English needs to be revised
Author Response
Thank you for submitting your review to molecules
Here are my few comments
The review article talked about the significance of nucleotides as secondary messengers within bacteria. The authors started with an overview of c-di-GMP functions concerning biofilm regulation and cell motility, highlighting its downstream effects on pili, adhesions, and polysaccharides. Then, they explained the distinct roles of cAMP signaling within various pathogenic bacteria and the contribution of c-di-AMP in recognizing DNA damage. Finally, The authors disclosed the interaction between alarmones and c-di-AMP, along with c-di-CMP.
Although this constitutes a review article and not a research-based manuscript containing results and discussions, it's still valuable. Prior literature, such as the articles found at the two provided links, have indeed explored the same title. Nevertheless, the addition of varied review articles within the same area of study continues to enrich the academic literature.
- https://www.sciencedirect.com/science/article/pii/S136952740900006X
- https://www.sciencedirect.com/science/article/pii/S0022283619300129
Answer:
The topic of bacterial biology is interesting; so much information is available. However, the first link is a review focusing more on c-di-GMP. The second link is similar to ours in general structure, but our review is more up-to-date to enrich this study area.
This manuscript simplifed some of the roles of nucleotides as secondary messengers in bacteria however, the authors were unable to elucidate the molecular mechanism of the environmental signal that triggers their synthesis and degradation.
the authors were unable to elucidate the molecular mechanism of the environmental signal that triggers the synthesis and degradation of these secondary messengers which is still a gap in the literature but yes, I can say that their conclusion is consistent with the argument presented.
Answer:
In the study of c-di-nucleotides, the molecular mechanism of activation by environmental changes and deactivation of this signaling process is unknown. In conclusion, we point out this part for the future study of c-di-nucleotide signaling.
As I mentioned in my first comment, having more than 100 references is too much, however the authors tried to avoid self-citation.
Answer:
In this new version of the manuscript, the English language was revised in detail by a native US researcher colleague, which improved the writing of the paper.
In this new version of the manuscript, English was revised.
Reviewer 2 Report
Comments and Suggestions for Authors
In this review, Cancino-Diaz et al. listed the nucleotides acting as second messengers in prokaryotic cells. They addressed several classes of c-di-nucleotides, highlighted the biological functions of cyclic dinucleotides in the signaling in bacteria. The work appears to be comprehensive, all references are well cited and the topic will receive wide interest by nucleotide signaling community. The new platform provides valuable insights into biological functions of nucleotides, offering implications for future studies and practical applications. However, I do have some questions/concerns that I think the authors should address before publication.
1. In the introduction section, the authors should give the chemical structure of nucleotides described in this review.
2. In the introduction part, the biogenesis of nucleotides should be illustrated in one figure.
3. The structure of paper is confusing and chaos. After introduction part, the first section is “1. Similarities and differences between c-di-nucleotides”, however, the next part number is still 1, the other nucleotide section number always is 1. And the subsection is always listed as “1.1”. In the sub-section of “The cAMP nucleotide”, the number is suddenly changed to 5.3. And it is not necessary to give another section “1.1.1. Cell wall homeostasis”.
4. In the types of c-di-GMP receptors section, the authors listed the receptor proteins of c-di-GMP in different prokaryotic cells. What’s the biological function of these proteins should be described.
5. Is there any clinical application of leveraging nucleotide signaling pathways. The author should use another section to describe how to use these pathways to control the life cycles of bacteria.
Comments on the Quality of English LanguageNo
Author Response
Review 2.
In this review, Cancino-Diaz et al. listed the nucleotides acting as second messengers in prokaryotic cells. They addressed several classes of c-di-nucleotides, highlighted the biological functions of cyclic dinucleotides in the signaling in bacteria. The work appears to be comprehensive, all references are well cited and the topic will receive wide interest by nucleotide signaling community. The new platform provides valuable insights into biological functions of nucleotides, offering implications for future studies and practical applications. However, I do have some questions/concerns that I think the authors should address before publication.
- In the introduction section, the authors should give the chemical structure of nucleotides described in this review.
Answer:
In this new version, a new figure with the chemical structures of nucleotides was added.
- In the introduction part, the biogenesis of nucleotides should be illustrated in one figure.
Answer:
Biogenesis and degradation are in Figures 3, 4, and 5, along with the nucleotide functions. These figures are referred to in the introduction.
- The structure of paper is confusing and chaos. After introduction part, the first section is “1. Similarities and differences between c-di-nucleotides”, however, the next part number is still 1, the other nucleotide section number always is 1. And the subsection is always listed as “1.1”. In the sub-section of “The cAMP nucleotide”, the number is suddenly changed to 5.3. And it is not necessary to give another section “1.1.1. Cell wall homeostasis”.
Answer:
Apologies for the poor organization of the manuscript. In this new version, the topics have been restructured adequately.
- In the types of c-di-GMP receptors section, the authors listed the receptor proteins of c-di-GMP in different prokaryotic cells. What’s the biological function of these proteins should be described.
Answer:
In the later paragraphs of the section "2.1. Types of c-di-GMP receptors," an example of the most common types of c-di-GMP receptors is discussed. Other receptors in specific functions are mentioned in the following sections.
- Is there any clinical application of leveraging nucleotide signaling pathways. The author should use another section to describe how to use these pathways to control the life cycles of bacteria.
Answer:
A new section (section 7) was added (Clinical application of di-nucleotides); please see the manuscript.
Round 2
Reviewer 2 Report
Comments and Suggestions for Authors
The authors solved my questions.
Comments on the Quality of English LanguageThe language in this manuscript could benefit from further refinement. I strongly recommend that the author seek the assistance of a native speaker to enhance the overall quality of the manuscript.